# Machine Learning to Understand the Immune-Inflammatory Pathways in Fibromyalgia

**DOI:** 10.3390/ijms20174231

**Published:** 2019-08-29

**Authors:** Laura Andrés-Rodríguez, Xavier Borràs, Albert Feliu-Soler, Adrián Pérez-Aranda, Antoni Rozadilla-Sacanell, Belén Arranz, Jesús Montero-Marin, Javier García-Campayo, Natalia Angarita-Osorio, Michael Maes, Juan V. Luciano

**Affiliations:** 1Group of Psychological Research in Fibromyalgia & Chronic Pain (AGORA), Institut de Recerca Sant Joan de Déu, 08950 Esplugues de Llobregat, Spain; 2Teaching, Research & Innovation Unit, Parc Sanitari Sant Joan de Déu, 08830 St. Boi de Llobregat, Spain; 3Primary Care Prevention and Health Promotion Research Network, RedIAPP, 28013 Madrid, Spain; 4Faculty of Psychology, Universitat Autònoma de Barcelona, 08193 Bellaterra (Cerdanyola del Vallès Barcelona, Spain; 5Department of Rheumatology, Parc Sanitari Sant Joan de Déu, 08830 St. Boi de Llobregat, Spain; 6Parc Sanitari Sant Joan de Déu, CIBERSAM, 08950 Sant Boi de llobregat, Spain; 7Department of Psychiatry, Miguel Servet Hospital, Aragon Institute of Health Sciences (I+CS), 50009 Zaragoza, Spain; 8Department of Psychiatry, Chulalongkorn University, Bangkok 10330, Thailand

**Keywords:** fibromyalgia, widespread pain, cytokines, inflammation, neuro-immune

## Abstract

Fibromyalgia (FM) is a chronic syndrome characterized by widespread musculoskeletal pain, and physical and emotional symptoms. Although its pathophysiology is largely unknown, immune-inflammatory pathways may be involved. We examined serum interleukin (IL)-6, high sensitivity C-reactive protein (hs-CRP), CXCL-8, and IL-10 in 67 female FM patients and 35 healthy women while adjusting for age, body mass index (BMI), and comorbid disorders. We scored the Fibromyalgia Severity Score, Widespread Pain Index (WPI), Symptom Severity Scale (SSS), Hospital Anxiety (HADS-A), and Depression Scale and the Perceived Stress Scale (PSS-10). Clinical rating scales were significantly higher in FM patients than in controls. After adjusting for covariates, IL-6, IL-10, and CXCL-8 were lower in FM than in HC, whereas hs-CRP did not show any difference. Binary regression analyses showed that the diagnosis FM was associated with lowered IL-10, quality of sleep, aerobic activities, and increased HADS-A and comorbidities. Neural networks showed that WPI was best predicted by quality of sleep, PSS-10, HADS-A, and the cytokines, while SSS was best predicted by PSS-10, HADS-A, and IL-10. Lowered levels of cytokines are associated with FM independently from confounders. Lowered IL-6 and IL-10 signaling may play a role in the pathophysiology of FM.

## 1. Introduction

Fibromyalgia (FM) is a chronic syndrome of unknown pathophysiology characterized by widespread musculoskeletal pain coupled with fatigue, stiffness, disordered sleep, perceived cognitive dysfunction, and mood disturbances [1]. One of the current hypotheses suggests that an unbalanced immune system may play a role in the pathophysiology of FM [2]. Several studies evaluated the role of interleukins (IL), chemokines, and immune-related substances in FM, resulting in inconsistent results. For example, increased levels of IL-6, CXCL-8, high-sensitivity C-reactive protein (hs-CRP), and IL-10 were found in some [3,4,5], but not all studies [6,7,8]. Moreover, other results supported the view that FM may be accompanied by suppression of the immune-inflammatory response system (IRS) [8]. In addition, in order to obtain a more comprehensive picture of immune function in FM, it is important to consider not only biomarkers of the IRS, including CXCL-8 and hs-CRP, but also immune regulatory or anti-inflammatory cytokines, including IL-10, and cytokines with combined IRS and regulatory properties such as IL-6 [9].

Results of published systematic reviews and meta-analyses indicate that methodological differences among immune studies are a main source of inconsistent results [10,11]. Major sources of heterogeneity might be due to uncontrolled variables including age, sex, body mass index (BMI) [12], sleep disorders [13,14], or comorbid medical disorders [15,16]. In addition, there is a wide range of clinical extraneous variables that were constantly disregarded, including depressive and anxiety symptoms [17,18], intake of antidepressants (ADs) [19,20], menopause [21], use of anticonvective hormones [22], aerobic exercise [23,24], seasonality, and diurnal variations [25,26].

Hence, in this case-control study we examine serum levels of pro-inflammatory markers, including CXCL-8 and hs-CRP, as well as IL-6 and the major negative immune-regulatory cytokine IL-10 in FM as compared with healthy controls, while adjusting for most of the above-mentioned extraneous variables. Based on the immune-inflammatory hypothesis of FM, the specific hypothesis is that - after controlling for relevant socio-demographic and biomedical confounders - FM is characterized by higher levels of IL-6, CXCL-8, and hs-CRP and lowered or unchanged levels of IL-10. Additionally, the predicting ability of the inflammatory biomarkers along with other predictors (e.g., severity of depression and anxiety) was examined using neuronal network (NN) machine learning. NN is well suited to discover more complex nonlinear relationships between input variables (i.e., cytokines, depression, anxiety) predicting FM diagnosis and widespread pain scores (the outcome variables) while examining multiple interrelated factors. Additionally, NN allows to rank the input variables in order of predictive power of the features.

## 2. Results

### 2.1. Socio-Demographic and Clinical Data in Patients and Controls

No significant differences were found between cases and controls in age, living arrangement, menopause, perceived healthy diet, or smoking. Patients showed significant lower education, employment status, and more use of medication, number of comorbidities, and a higher BMI, as compared with controls. The WPI, SSS, and FS scores were significantly higher in FM than in controls. The severity of FM as assessed with FIQ-R in FM patients was moderate [27], and they suffered from FM for 13.0 (±9.5) years. All clinical rating scale scores were significantly different between the two groups. Table 1 shows the raw data of the 4 biomarkers in both study groups, while Electronic Appendix A and Appendix A illustrate the biomarker differences between the study groups. Nevertheless, we would refer the reader to Table 2 and Table 3 which computes the estimated marginal means of the immune biomarker data after adjustment for the relevant confounders.

### 2.2. Associations Between Cytokines and Diagnosis

Table 2 shows the results of multivariate multivariate general linear model (GLM) analyses with the cytokines and the composite score as dependent variables and significant confounders, namely age, BMI groups, and quality of sleep as explanatory variables. The other extraneous variables were not significant in this analysis, including comorbidities, intake of antidepressants and other medications, aerobic activity, sleep measurements, and clinical rating scale scores. Before adjusting for the significant covariates, we found that there was a significant between-group difference in immune markers and the composite score. Tests for between-subjects effects showed significant effects of diagnosis (see also Table 1). Nevertheless, after covarying for age, quality of sleep, and BMI groups in the same multivariate GLM analysis, we found that the effects of diagnosis on these 5 dependent variables were no longer significant. Tests for between subjects (see Table 2) showed that there were still significant differences between cases and controls in IL-6, CXCL-8 and IL-10 and their composite score, whereas the effects on hs-CRP were no longer significant. P-correction for FDR shows that IL-6 (*p* = 0.04), CXCL-8 (*p* = 0.05), IL-10 (*p* = 0.02) and their composite score (*p* = 0.02) remained significant. There was a significant effect of BMI-groups on hs-CRP whereby the latter increased from the BMI group with <25 kg/m^2^ (mean ±SD: −0.47 ± 0.18 in z scores) to 25–30 kg/m^2^ (0.23 ± 0.17) to the group with >30 kg/m^2^ (0.35 ±0.20). There was a significant inverse association between IL-6 and age, while quality of sleep showed a very modest inverse association with IL-10. 

Table 3 shows the mean values of the five immune markers in z scores after adjusting for the confounders. IL-6, CXCL-8, IL-10, and the composite score were significantly lower in FM as compared with controls, while there were no differences in hs-CRP. For example, the difference in IL-10 between both study groups showed that IL-10 was 0.70 SDs lower in FM than in HC. There were no significant effects of the use of ADs on the five immune biomarkers, either in a multivariate GLM analysis (F = 1.07, df = 4/91, *p* = 0.38) or tests for between-subject effects (all *p* > 0.22 even without *p*-correction for FDR). There was no significant effect of the use of pain killers on the same biomarkers. There were no significant differences in the five biomarkers between a) subjects with and without a current episode of major depression either using multivariate GLM (F = 0.15, df = 4/57, *p* = 0.96) or tests for between-subjects effects (all *p* > 0.52), and b) subjects with and without a lifetime episode of major depression either using multivariate GLM (F = 1.35, df = 4/57, *p* = 0.27) or tests for between-subjects effects (all *p* > 0.08). There were no significant effects of smoking (yes/no) on the five immune biomarkers, either in a multivariate GLM analysis (F = 0.77, df = 4/87, *p* = 0.55) or tests for between-subject effects (all *p* > 0.20 even without *p*-correction for FDR).

### 2.3. Prediction of FM

In order to check the best predictors of FM as dependent variable (with controls as reference group) we performed logistic regression analyses. We entered the cytokines as explanatory variables while possible confounders were comorbidities, quality of sleep, aerobic exercise, HADS-A, HADS-D, and PSS-10. Adding smoking as a covariate (forced entry) showed that there was no significant effect of smoking on the outcome of these regression analyses. The first logistic regression analysis shown in Table 4 indicates that FM was best predicted by IL-10 even after controlling for quality of sleep, aerobic exercise (all inversely) and comorbidities (positively) (*χ^2^* = 63.32, df = 4, *p* < 0.001, Nagelkerke = 0.64). We performed a second regression analysis without comorbidities as explanatory variable and found that FM was predicted by IL-10 even after adjusting for quality of sleep (both inversely) and HADS-A (*χ^2^* = 52.38, df = 3, *p* < 0.001, Nagelkerke = 0.56). These findings show that IL-10 is a better predictor of FM than IL-6 and CXCL-8 and that the inverse association between IL-10 and FM remains significant after adjusting for quality of sleep, HADS-A, and aerobic exercise.

### 2.4. Results of NN

NN was employed to predict the FS rating subscales scores (WPI and SSS scores) as well as the FM diagnosis using 11 predictors as input variables, namely the 4 immune biomarkers, HADS-D, HADS-A, PSS-10, BMI-groups (split according to a BMI of 25.0 kg/m^2^), comorbidities, quality of sleep, and aerobic activity. The summaries of the automatic architecture training of the network models are shown in Table 5.

NN model #1 (with WPI as output variable) was trained with two hidden layers with 4 units in hidden layer 1 and 3 units in hidden layer 2. Both layers employed hyperbolic tangent as activation function, while the output layer used identity as activation function. This neural network used the sum of squares as error term. Table 5, NN model #1 shows the model summary data. During the training, model #1 minimized the error term, indicating that the model learnt to generalize from the trend. The relative error terms were fairly constant among the training and either testing or holdout sample indicating that the trained models were not overtrained. The correlation coefficient between the WPI score and the model-predicted value was *r* = 0.60. Figure 1 shows the relevance chart with importance and normalized importance of the 11 input variables. Quality of sleep, PSS-10, and HADS-A were the most important determinants of the predictive power of the network followed by IL-6, CXCL-8, and IL-10.

In NN model #2 the SSS score was the output variable while the same 11 input variable without HADS-D and quality of sleep (because those symptoms are part of the SSS score) were introduced. This model was trained with one hidden layer with 4 units employing hyperbolic tangent as activation function, while the output layer used identity as activation function. This model minimized the error term, indicating that the model learned to generalize from the trend. The relative error terms were fairly constant among the three samples and the correlation coefficient between the SSS score and the model-predicted value was r = 0.63. Figure 2 shows the relevance chart with importance and normalized importance of the 9 input variables. PSS-10, HADS-A, and IL-10 were the most important determinants of the predictive power of NN model #2, followed at a distance by comorbidities and IL-6.

In NN model #3 the output variables were FM diagnosis and the control group with again the same 11 input variables. Table 5, model #3 shows that the percentage of incorrectly classified cases was relatively constant among the training, the testing or holdout samples. Figure 3 shows that the area under the receiving operating curve (ROC) was 0.91. In the holdout set, we found that 87.1% of all cases were correctly classified by the model. Figure 4 shows the relevance chart for the discrimination of FM patients versus HC. Quality of sleep, HADS-A, aerobic activity, and PSS-10 were the most important determinants of the predictive power of neural network 3, followed at a distance by BMI groups, CXCL-8, IL-10, and comorbidities.

## 3. Discussion

The major finding of this case-control study is that, contrary to our a priori hypothesis, IL-6 and CXCL-8 levels were significantly decreased in patients with FM compared with controls. Moreover, IL-10 was also significantly lower in FM patients, while there were no significant differences in hs-CRP after controlling for confounding variables. Our results on IL-6 levels are not in agreement with most published papers, which reported higher levels of IL-6 in FM [6,28,29,30], although some studies found no significant differences between FM patients and controls [4,8,31,32,33,34]. Previously, some authors reported lower levels of CXCL-8 in FM [28,30]. Nevertheless, as with IL-6, there is a large heterogeneity between published studies with a number of studies reporting no significant differences in CXCL-8 [3,33,35] and some even increased CXCL-8 levels [30]. In accordance with a previous study, we found lowered levels of IL-10 in FM [6], although other studies reported higher levels of IL-10 [3,4]. Similarly, previous results on hs-CRP in FM were mixed with some studies reporting higher levels [34,36], while other studies could not find any differences between FM patients and controls [37,38].

All in all, the results show that there is a large heterogeneity among studies in cytokine-chemokine results. This is further corroborated by the results of a recent systematic review [10], which was unable to delineate a specific immune profile in FM and concluded that the methodological quality of immune studies in FM should be improved. A major source of heterogeneity is that most studies did not control or only partially controlled for the potential important confounders, which may modulate immune biomarkers. In the present study, we found that after controlling for BMI, comorbid illnesses, aerobic exercise, anxiety, depression, perceived stress symptoms, and quality of sleep, the differences in IL-6, IL-10, and CXCL-8 between groups remained significant, whereas hs-CRP was no longer significantly increased in FM. Since patients with FM usually present with higher BMI values than observed in the general population [39,40], it is plausible that the differences in hs-CRP levels reported in some studies are attributable to effects of BMI. Previously, it was shown that BMI is also a major determinant of increased hs-CRP values in bipolar disorder and major depression [41,42]. In this work, we have found that two other confounders, namely age and quality of sleep, may interfere with cytokine-chemokine measurements in FM. Firstly, age was significantly and negatively correlated with IL-6 levels, although some but not all studies showed that IL-6 may increase with age [43]. Secondly, we found a very modest inverse association between quality of sleep and IL-10 levels. Moreover, FM is a syndrome with a high prevalence of comorbidities with medical and psychiatric conditions (e.g., obesity, arthrosis, depression, anxiety), which can influence peripheral levels of immune-inflammatory biomarkers and which should, therefore, be considered when interpreting cytokine results. As such, the effects of comorbidities combined with other variables including quality of sleep, BMI, age, sex, and exercise could explain the discrepant results reported in the literature.

Another important issue is that it is difficult to interpret results on cytokines-chemokines without considering their pro- versus anti-inflammatory profile, because the balance between pro-/anti-inflammatory biomarkers determines the functional state of the immune system [44]. As such, we have assayed 2 pro-inflammatory mediators (CXCL-8 and hs-CRP), the major negative immune-regulatory cytokine (IL-10) and a cytokine (IL-6) that may exhibit pro-inflammatory as well as anti-inflammatory effects [44,45]. In this regard, z unit weighted composite scores may be computed which denote different functional immune ratios including pro/anti-inflammatory or IRS/regulatory [44]. In our study, we expected to find immune activation as characterized by an immune profile with increased levels of the four biomarkers or an inflammatory response as indicated by increased levels of hs-CRP, IL-6 and CXCL-8, and possibly a relative decrease in IL-10, an anti-inflammatory and regulatory cytokine [44]. Nevertheless, the immune profile observed in our study showed a reduction in IL-6, IL-10, and CXCL-8 indicating that there is no clear immune activation and no inflammatory response (according to these classic biomarkers) in our FM patients. This is important as our findings do not corroborate the major hypothesis that IL-6 and other cytokines, which may induce pain, mental, and cognitive impairments, are increased in that illness. Recently, IL-6 has emerged as a key regulator of pathologic pain and administration of IL-6 may cause allodynia and hyperalgesia [46]. Pain models are frequently accompanied by increased levels of IL-6 and the IL-6 receptor (IL-6R) in the spinal cord and dorsal root ganglia [46]. This hyperalgesic effect of IL-6 is probably mediated via IL-6 trans-signaling through increased levels of the soluble IL-6 receptor (sIL-6R), which binds with IL-6 to form an active pro-inflammatory complex [47].

The second major finding of our study is that regression analysis showed that, taking into consideration the measured immune biomarkers, IL-10 is the best predictor of FM diagnosis, while NN analysis showed that the three cytokines measured herein are the best predictors of the WPI score and IL-10 the best immune biomarker predicting the SSS score. Comparable results indicating lowered levels of anti-inflammatory cytokines were reported in a study that examined IL-4 and IL-10 gene expression patterns and serum concentrations in chronic widespread pain patients including FM [48]. IL-4 is a T helper-2 (Th-2) cytokine, and IL-10 is a T regulatory (Treg) cytokine, and both may suppress an immune response [9]. Thus, these authors found lower levels of the anti-inflammatory cytokines IL-4 and IL-10 in patients with chronic pain, whereas there were no significant differences in pro-inflammatory cytokines-chemokines including IL-2, CXCL-8, tumor necrosis factor (TNF)-α, and transforming growth factor (TGF)-β1, another immune-regulatory cytokine. The authors concluded that chronic pain is associated with lowered levels of anti-inflammatory and analgesic Th-2 (and Treg) cytokine activity [48]. Both Th-2 and Treg cytokines attenuate the production of pro-inflammatory M1 macrophage and Th-1 cytokines, suggesting that lower levels of IL-10 in FM may attenuate the negative feedback on the inflammatory response system. Nevertheless, in our study some other immune markers were suppressed (IL-6 and CXCL-8) or unchanged (hs-CRP), while in the study of Üçeyler et al. [48] reduced Th-2 and Treg activities were not accompanied by signs of immune activation in patients with chronic pain.

The above findings show that there should be another explanatory hypothesis, which combines the effects of lowered IL-10 and IL-6 in explaining FM symptoms. One hypothesis is that this immune profile may indicate loss of neuroprotection because IL-10 [49] and classical IL-6 signaling [45,50] have (neuro)protective properties. For example, the lack of IL-10 may accelerate lesion expansion following excitotoxic injuries in association with the onset of pain behaviors, while IL-10 may protect against this type of lesions and the accompanying pain [51]. Furthermore, FM is accompanied by highly increased serum levels of sgp130, the natural inhibitor of IL-6 trans-signaling and an inhibitor of the activity of leukemia inhibitor factor (LIF) and oncostatin [52]. LIF is a growth factor that plays a role in genomic plasticity; it has regenerative effects and promotes T regulatory functions, and repair and tolerance in the central nervous system [53,54]. Oncostatin is produced by myofibers and maintains the regenerative capacity of stem cells in the muscle [55]. As such, it appears that FM may be accompanied by disorders in the IL-6–sgp130 system and in regenerative IL-6-related cytokines.

Last but not least, our neural networks analysis showed that widespread pain symptoms (WPI subscale score) are strongly predicted by sleep quality, perceived stress and anxiety, and that the FM diagnosis is significantly predicted by sleep quality, perceived stress, anxiety, and aerobic activities indicating that FM is much more than a pain condition. These results extend the findings of Wolfe et al. [56] who reported that the original FM criteria (1990) [57] underestimated the prevalence of FM by approximately 25% and that therefore it is more appropriate to use the revised ACR diagnostic criteria, which comprise sleep problems, depression, fatigue, gastro-intestinal, and cognitive symptoms.

Our findings should be interpreted with regard to its limitations. Firstly, this study recruited only females who represent the majority, but not all patients with FM. According to the last ACR criteria, there are around 33% males in the FM population. Secondly, the sample size was rather small particularly the normal control group. Therefore, our findings deserve replication in males using a larger study sample. Thirdly, future research should include of a more comprehensive list of cytokines, including M1 macrophage, Th-1, Th-2 and Th-17 cytokines altogether with other proteins implicated in the immune status, such as sgp130, LIF, and oncostatin.

## 4. Materials and Methods

### 4.1. Participants

The present study had a cross-sectional case-control design embedded within a full randomized controlled trial (N = 225) [58]. Sixty-seven female patients with FM were recruited at Sant Joan de Déu Hospital (St. Boi de Llobregat, Spain) and 35 healthy female age-matched controls were contacted from the hospital staff and acquaintances. The present work was performed in accordance with the ethical standards laid down in the 1964 Declaration of Helsinki and its subsequent updates. The Fundació Sant Joan de Déu (FSJD) Research Committee Board evaluated and approved the study protocol (PIC-102–15, 12 November 2015).

The inclusion criteria for FM patients were: women, aged 18 to 65 years, FM diagnosis according to American College of Rheumatology (ACR) 1990 criteria [57], understanding of Spanish, and give written informed consent. The inclusion criteria for controls were: women, age-matched with FM patients, absence of current or past FM diagnosis confirmed with the Fibromyalgia Symptoms Scale (FS) [56], understanding of Spanish, and give written informed consent. Exclusion criteria for both patients and controls were: recent injury/trauma, (auto-)immune and neuro-inflammatory diseases, major psychiatric disorders (e.g., schizophrenia, psycho-organic syndromes), common cold or flu during the day of blood extraction, needles’ phobia, BMI ≥36 kg/m^2^ or weight ≥110 kg, ≥8 caffeine/units per day, smoker of ≥5 cigarettes per day, pregnancy or breastfeeding, use of local or oral corticosteroids, use of immune modulatory drugs, and use of oral contraceptives.

### 4.2. Clinical Measurements

Clinical evaluation and sociodemographic characteristics were assessed by a health psychologist. A comprehensive battery of self-reported scales was administered [58]. The Spanish version of the Revised Fibromyalgia Impact Questionnaire (FIQ-R) [59,60] was administered to the FM patients in order to assess FM severity. This scale includes 21 items on an 11-point numeric rating scale from 0 to 10. Time frame is the previous 7 days and the items are distributed into three associated domains: physical function, overall impact, and severity of symptoms. Total scores can range from 0 to 100, with higher scores indicating more functional impairment.

The Fibromyalgia Symptoms Score (FS) [56,61] is a 6-item self-report questionnaire that registers the key symptoms of FM according to the latest revision of the ACR criteria. It includes assessment through the Widespread Pain Index (WPI), with a total score ranging from 0 to 19, and the Symptom Severity Scale (SSS), which ranges from 0 to 12. It includes fatigue, trouble thinking/remembering, and waking up tired or unrefreshed, pain or cramps in the lower abdomen, depression, and headache. A total FS score of “fibromyalgia-ness” can be computed by summing WPI items and the SS scale (ranging from 0–31).

The Hospital Anxiety and Depression Scale (HADS) [62,63,64] includes a total of 14 items divided in two sub-scales of 7 items each, assessing anxiety (HADS-A) and depression (HADS-D) symptoms. Scores range from 0 to 21 for both sub-scales. The general distress total score ranges from 0 to 42, with higher scores indicating greater severity.

The Perceived Stress Scale (PSS-10) [65] is a 10 item scale that measures self-reported stress level in the past month, with scores ranging from 0 to 40 being higher scores indicative of greater perceived stress.

Using a semi-structured interview and visual analogue scales we also assessed health-related variables including sleep quality (rated as 0 “poor quality of sleep” to 10 “high quality of sleep”), healthy diet (rated 0 “very unhealthy diet” to 10 “very healthy diet”), menopause (yes/no), use of recreational drugs (yes/no), smoking (yes/no), practice of aerobic exercise (yes/no), and use of antiinflammatories, analgesics, antieplieptics, opioids, anxiolitics, and oral contraceptives [12]. Additionally, we computed a comorbidity index based on number of present chronic comorbid conditions including cardiovascular, endocrine, metabolic, neurodegenerative, rheumatological and respiratory comorbidities. Body mass index (BMI) was calculated as body weight [kg]/height [m^2^].

### 4.3. Blood Assays

Blood samples were collected between 8.00–10.00 am. After blood extraction, samples were centrifuged, and serum was separated and frozen at −80 °C until thawed for biomarkers assays, which were performed in one and the same run. We used the Milliplex^®^ Multiplex Assay (MerckMillipore^®^ Burlington, MA, USA) in a Luminex^®^ platform to measure IL-6 (normal range 1–2 pg/mL; detection limit: 0.11 pg/mL), CXCL-8 (normal range 1–157 pg/mL; detection limit: 0.12 pg/mL) and IL-10 (normal range 4–17 pg/mL; detection limit: 0.51 pg/mL) (Human High Sensitivity T Cell Panel, catalog n HSTCMAG-28SPMX adapted to the three mentioned cytokines). Hs-CRP (detection limit ≤1 mg/L) was analyzed by turbidimetry in an Olympus AU5400 Autoanalyser (Olympus Iberia S.A.U., l’Hospitalet de Llobregat, Spain).

### 4.4. Statistics

We used mean ± standard deviations (SDs) to represent continuous variables and percentages for categorical variables. Biomarker data were first Ln transformed (IL-6, CXCL-8 and hs-CRP) and consequently z transformed. For between-group comparisons, we computed one-way analyses of variance for continuous values and χ^2^ tests for categorical values. Pearson’s product moment correlation or Spearman’s rank order correlation coefficients were used to compute associations between scale variables. Multivariate general linear model (GLM) analysis was used to assess the effects of diagnosis on immune markers, while controlling for confounders including age, smoking, and BMI groups, namely subjects with BMI <25.0 kg/m^2^, between 25–30 kg/m^2^ and >30 kg/m^2^. Consequently, tests for between-subject effects were carried out to assess the effects of diagnosis on each of the immune markers. We computed model-generated estimated marginal mean values and displayed differences in mean z (±SE) values among diagnostic categories. A composite score integrating the cytokines/chemokines was computed summing the values of z IL6 + z IL10 + z CXCL-8 [44]. All results of multiple tests were *p*-corrected for false discovery rate (FDR) [64]. In addition, we employed stepwise binary logistic regression analysis to assess the significant explanatory variables predicting FM diagnosis (yes/no) as dependent variable and controls as reference group, and we computed Odds ratios with 95% confidence intervals (CI), while Nagelkerke values were used as effects size estimates. All analyses were bootstrapped (*n* = 1000), and these results are reported when differences between this approach and the non-bootstrapped results were observed. All the tests were two-tailed and a *p*-value of 0.05 was used for statistical significance.

Multilayer perceptron neural network (NN) analyses were used to examine the more complex associations between the FM diagnosis, WPI, and SSS scores (output variables), and immune markers, age, BMI, smoking, and clinical rating scales as input variables in automated feedforward architecture models. The networks were trained using one or two hidden layers with a variable number of nodes (up to 7) with batch (gradient descent) and minibatch (automatically determined number of records). As stopping rule, we used one consecutive step with no further decrease in the error term. The maximum number of training epochs was set at 200, while the network weights were randomly initialized. Part of the cases (46.6%) were assigned to the training set (to estimate the network parameters), 20.0% to the testing set (to prevent overtraining) and 33.3% to the holdout set (to evaluate the final network). We computed error, relative error and importance of each of the input variables in sensitivity analyses. The importance (or relative importance) reflects the strength of the relationship between the input variables and their multiple interactions and the output variable, and as such indicates the predictor importance. These predictor importance values are relative because they are computed yielding a sum of 1 or 100%. These relative values are computed based on the weights that describe the input variables as well as their multiple interactions in the neural network (SPSS, 2019 [65]). All statistical analyses were performed using IBM SPSS windows version 25 (IBM, Armonk, NY, USA).

## 5. Conclusions

In our FM patients, we found lowered levels of IL-6, IL-10, and CXCL-8, independently from the effects of confounder variables, whereas hs-CRP was not altered in FM. IL-10 is the best immune biomarker predicting FM diagnosis. The severity of widespread pain is best predicted by quality of sleep, perceived stress, anxiety, and the three cytokines, while severity of FM is best predicted by stress, anxiety, and IL-10.

## Figures and Tables

**Figure 1 ijms-20-04231-f001:**
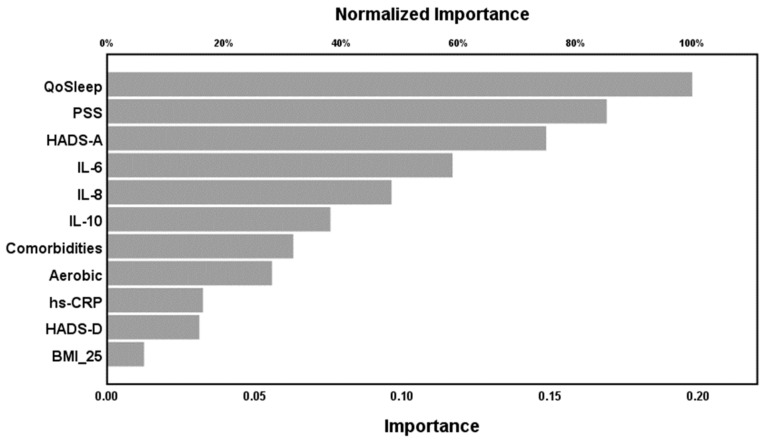
Neural network model #1 with WPI as output. QoSleep: Quality of Sleep, PSS(-10): Perceived Stress Scale, HADS-A: Anxiety sub-scale of the Hospital Anxiety and Depression Scale, IL: interleukin, Comorbidities: number of total comorbid illnesses, Aerobic: aerobic exercise, times per week, hs-CRP: high sensitivity C-reactive protein, HADS-D: Depression sub-scale of the Hospital Anxiety and Depression Scale, BMI_25: Body Mass Index >25 kg/m^2^.

**Figure 2 ijms-20-04231-f002:**
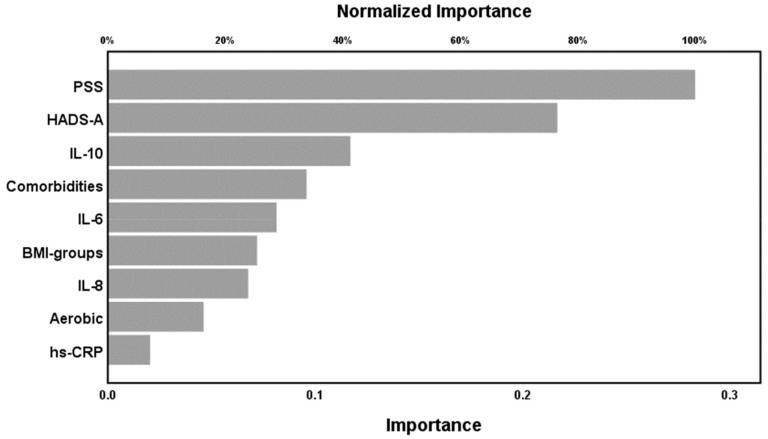
Neural network model #2 with SSS as output. PSS(-10): Perceived Stress Scale, HADS-A: Anxiety sub-scale of the Hospital Anxiety and Depression Scale, IL: interleukin, Comorbidities: number of total comorbid illnesses, BMI-groups: Body Mass Index categorical variable, Aerobic: aerobic exercise, times per week, hs-CRP: high sensitivity C-reactive protein.

**Figure 3 ijms-20-04231-f003:**
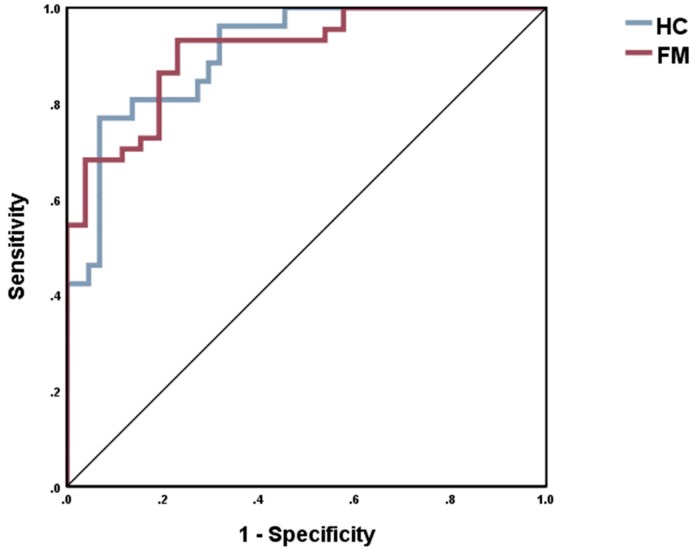
Area under the receiving operating curve (0.91) of the Neural Networking model #3 with FM diagnose as output.

**Figure 4 ijms-20-04231-f004:**
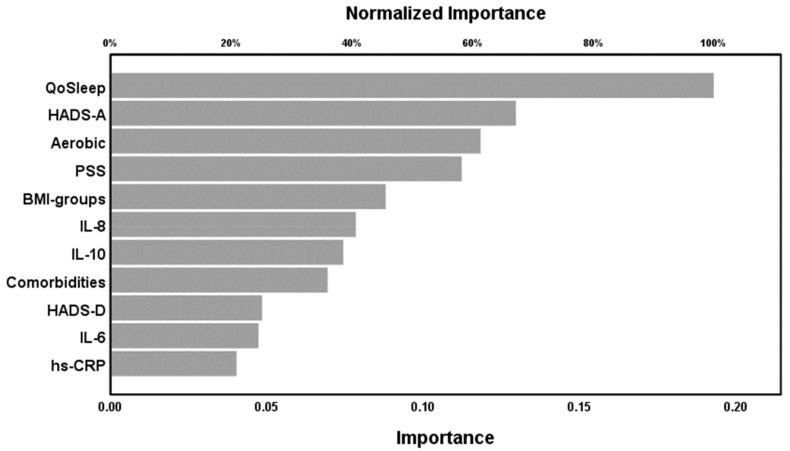
Neural network model #3 with FM diagnose as output. QoSleep: Quality of Sleep, HADS-A: Anxiety sub-scale of the Hospital Anxiety and Depression Scale, Aerobic: aerobic exercise, times per week, PSS-10: Perceived Stress Scale, BMI: Body Mass Index, IL: interleukin, Comorbidities: number of total comorbid illnesses, HADS-D: Depression sub-scale of the Hospital Anxiety and Depression Scale, hs-CRP: high sensitivity C-reactive protein.

**Table 1 ijms-20-04231-t001:** Socio-demographic, clinical and biomarker data in patients with fibromyalgia (FM) and healthy controls (HC).

Variables	FM (*n* = 66)	HC (*n* = 35)	*p*
Sociodemographic & biometric variables			
Age (years)	53.36 (8.05)	54.43 (7.84)	0.55
Employed (*n*, %)	26 (38.40)	26 (74.30%)	<0.001 *
With primary studies or more (*n*, %)	62 (94.00%)	35 (100.00%)	<0.001 *
Living with someone (*n*, % Yes)	61 (90.10%)	30 (85.70%)	0.23
BMI (kg/m^2^)	28.79 (5.27)	24.03 (3.67)	<0.001 *
Duration of illness (years)	12.98 (9.48)	*na*	*na*
In menopause (*n*, % Yes)	33 (55.00%)	22 (62.90%)	0.72
Number of comorbidities	0.97 (1.24)	0.26 (0.51)	<0.001 *
Antidepressants intake (*n*, % Yes)	25 (37.90%)	1 (2.90%)	<0.001 *
Clinical variables			
FIQ-R (0–100)	59.51 (21.06)		
WPI (0–19)	13.5 (3.70)	1.6 (1.80)	<0.001 *
SSS (0–12)	8.4 (2.60)	2.1 (1.80)	<0.001 *
FS (0–31)	22.00 (5.50)	3.6 (3.20)	<0.001 *
HADS_A (0–21)	10.55 (4.41)	6.17 (3.20)	<0.001 *
HADS_D (0–21)	7.77 (5.06)	3.14 (3.33)	<0.001 *
HADS Total (0–42)	18.32 (8.17)	9.20 (6.07)	<0.001 *
PSS-10 (0–40)	22.00 (9.47)	12.57 (6.49)	<0.001 *
Health behaviours			
Sleep quality (0–10)	3.79 (2.07)	6.43 (1.61)	<0.001 *
Healthy diet (0–10)	6.84 (2.01)	6.52 (2.25)	0.24
Smoke (*n*, % Yes)	12.00 (19.70%)	5.00 (14.30%)	0.36
Practice of aerobic exercise (*n*, % Yes)	37.00 (60.70%)	32.00 (91.40%)	0.001 *
Immune biomarkers			
IL-6 (pg/mL)*	5.93 (9.27)	7.56 (5.04)	0.007 *
IL-10 (pg/mL)*	10.29 (10.78)	17.94 (15.02)	0.04 *
CXCL8 (pg/mL)*	11.68 (8.90)	14.39 (10.29)	0.04 *
hs-CRP (mg/L)*	2.69 (2.96)	1.33 (1.76)	0.005 *

Values are shown as means ±SD or *n* and percentages (%) where appropriate. All results of analyses of variance or analyses of contingency tables. *Immune biomarkers were processed in Ln transformation. BMI: body mass index, FIQ-R: Revised Fibromyalgia Impact Questionnaire; FS: Fibromyalgia Symptoms scale; HADS: Hospital Anxiety and Depression Scale; PSS-10: Perceived Stress Scale; IL: interleukin; hs-CRP: high sensitivity C-reactive protein; SSS: Symptom Severity Scale from the FS; WPI: Widespread Pain Index from the FS. *na*: no applicable.

**Table 2 ijms-20-04231-t002:** Results of multivariate GLM analysis with the biomarkers as the dependent variables and diagnosis as explanatory variable while adjusting for extraneous variables.

Tests	Dependent Variables	Explanatory Variables	F	df	*p*
Multivariate	4 cytokines + 1 composite	Diagnosis	3.50	4/96	0.010 *
Between-subject effects	IL-6	Diagnosis	7.54	1/99	0.007 *
	CXCL-8	Diagnosis	4.59	1/99	0.035 *
	IL-10	Diagnosis	5.00	1/99	0.028 *
	hs-CRP	Diagnosis	8.44	1/99	0.005 *
	zIL6 + zIL10 + zCXCL-8	Diagnosis	8.74	1/99	0.004 *
Multivariate	4 cytokines + 1 composite	Diagnosis	2.20	4/92	0.075
		Age	2.81	4/92	0.030 *
		QoSleep	1.81	4/92	0.134
		BMI groups	1.81	8/186	0.077
Between-subject effects	IL-6	Diagnosis	5.45	1/95	0.022 *
	CXCL-8	Diagnosis	4.40	1/95	0.039 *
	IL-10	Diagnosis	7.11	1/95	0.009 *
	hs-CRP	Diagnosis	0.70	1/95	0.406
	zIL6 + zIL10 + zCXCL-8	Diagnosis	8.78	1/95	0.004 *
	hs-CRP	BMI groups	6.46	1/95	0.002 *
	IL-10	QoSleep	4.40	1/95	0.039 *
	IL-6	Age	8.51	1/95	0.004 *

Diagnosis: Fibromyalgia versus healthy controls. IL: interleukin; hs-CRP: high sensitivity C-reactive protein. zIL6 + zIL10 + zCXCL-8: composite score computed as the sum of z values of these biomarkers; QoSleep: Quality of Sleep. BMI groups: <25 kg/m^2^, 25–30 kg/m^2^, >30 kg/m^2^. Note:Immune biomarkers were processed in Ln transformation.

**Table 3 ijms-20-04231-t003:** Model-generated estimated marginal means of the immune biomarkers obtained by GLM analysis shown in Table 2 in healthy controls (HC) and patients with fibromyalgia syndrome (FMS).

Variables	HC	FMS
zIL-6	0.370 (0.196)	−0.218 (0.124)
zCXCL-8	0.338 (0.205)	−0.215 (0.130)
zIL-10	0.446 (0.204)	−0.253 (0.129)
zhs-CRP	−0.069 (0.190)	0.135 (0.120)
zIL6 + zIL10 + zCXCL-8	0.472 (0.198)	−0.281 (0.125)

Note: IL: interleukin; hs-CRP: high sensitivity C-reactive protein. zIL6 + zIL10 + zCXCL-8: composite score computed as the sum of the z values of these biomarkers. Comparing FM patients who did not take any medication and HC, the between-group differences disappeared and only CXCL-8 was found lower in FM.

**Table 4 ijms-20-04231-t004:** Results of binary logistic regression analysis with FM group membership as dependent variable (with healthy controls as reference group) and the immune biomarkers as explanatory variables.

No	Explanatory Variables	B	SE	Wald	df	*p*	OR	95% CI
#1	IL-10	−0.710	0.337	4.447	1	0.035	0.492	0.254–0.951
	Comorbidities	1.302	0.653	3.976	1	0.046	3.676	1.022–13.218
	QoSleep	−1.020	0.236	18.720	1	<0.001	0.360	0.227–0.572
	Aerobic exercise	−2.738	0.956	8.201	1	0.004	0.065	0.010–0.421
#2	IL-10	−0.742	0.325	5.211	1	0.022	0.476	0.252–0.900
	HADS-A	0.179	0.080	5.014	1	0.025	1.196	1.023–1.399
	QoSleep	−0.816	0.203	16.157	1	<0.001	0.442	0.297–0.658

OR: Odds ratio, CI: confidence intervals. All variables were entered as z scores. IL: Interleukin; QoSleep: Quality of sleep; HADS-A: Anxiety sub-scale of the Hospital Anxiety and Depression Scale.

**Table 5 ijms-20-04231-t005:** Multilayer Perceptron (MLP) Neural Network (NN) models with the Fibromyalgia Symptoms Scale (FS) sub-scores or the diagnosis Fibromyalgia as output variables.

	Models	NN#1	NN#2	NN#3
Input Layer	Number of units	11 + bias	9	11 + bias
	Rescaling method	Standardized	Standardized	Adjusted normalized
Hidden layers	Number of hidden layers	2	1	2
	Number of units in hidden layer 1	4	4	3
	Number of units in hidden layer 2	3	-	2
	Activation Function	Hyperbolic tangent	Hyperbolic tangent	Hyperbolic tangent
Output layer	Dependent variables	WPI score	SSS score	FM *vs.* controls (group membership)
	Number of units	1	1	2
	Rescaling method	Standardized	Standardized	
	Activation function	Identity	Identity	Identity
	Error function	Sum of squares (SS)	Sum of squares (SS)	SS
Training	SS error	10.742	10.096	6.971
	Relative error or % incorrect	0.413	0.439	18.7%
	Prediction (sensitivity - specificity)			82.1%–80.0%
	SS error	4.959	3.857	1.612
Testing	Relative error or % incorrect	0.421	0.378	9.1%
	Prediction (sensitivity - specificity)			93.3%–83.3%
	AUC ROC			0.908
Holdout	Relative error or % incorrect	0.512	0.485	12.9%
	Prediction (sensitivity-specificity)			86.4%–88.9%
	Correlation with predicted values	*R* = 0.598	*R* = 0.625	

**Note:** SSS: Symptom Symptoms Scale from the FS; WPI: Widespread Pain Index from the FS.

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
