# Peer review of "Machine Learning to Understand the Immune-Inflammatory Pathways in Fibromyalgia"

_ijms, 2019, doi:10.3390/ijms20174231_

Round 1
Reviewer 1 Report
The manuscript by Andres-Rodriguez at al., is concentrated on establishing the levels of certain cytokines/chemokines, while adjusting for multiple other parameters like age, BMI, comorbid disorders, quality of sleep and others in fibromyalgia (FM). The strength of this work is that the data is presented not only as correlation of one parameter to the other, but also in more complex network applying regression analysis and the method of deep learning. The data is presented clear and the results are discussed with the appropriate literature. Moreover, the authors acknowledge the limitations of their study (group size and only females taken in to consideration) and point at future research to be performed regarding FM.
Even though the authors present the data quite clear throughout the whole manuscript, it could be beneficial to present the cytokine data from table 1 in a graph with individual plotted values, showing the differences between the patients and healthy controls.
Author Response
Dear Reviewer 1,
Thank you for your review, as requested, we have added 1 graph and 3 plots in an electronic supplementary file to illustrate the differences between patients with fibromyalgia and healthy controls.
Reviewer 2 Report
some issues/concerns....
More detailed explanations why so much heterogeneity in FM studies with respect to the inflammatory markers Il-6 Il-10 ?
Is smoking/not smoking relationship observed in current markers similar to other observations with these markers and smoking?
Fig 3. Please list SUC value as part of the name on the legend for each item.
Column widths in Table 2 needs to be properly adjusted. To reduce vertical spacing. Suggest decrease with of columns 1, 2,3, and column 4.
Please provide in, simple terms, the mathematical basis against which all final scores are produced. (this might help the understanding of the methodology for the non-machine learning readers).
In figs 1,2 & 4 the x axis "importance" : pleases clarify....does this represent the weighting in the final score?
Line 400 is too strong considering all the discussion of variation in observations of previous reports.
Please consider adding to the beginning of the first sentence.... "In the observed patients, ".
Also, please add a small statement on what a neural network is and why the use of the neural network makes sense.
Author Response
Dear Reviewer,
Thank you very much for reviewing our manuscript. The answers to your questions follow:
More detailed explanations why so much heterogeneity in FM studies with respect to the inflammatory markers Il-6 Il-10?
#ANSWER: Immune biomarkers are subjected to the intervening effects of multiple variables such as those reported in the Introduction. In the Discussion we emphasize in this regard the importance of variables such as age, sex, BMI, comorbid medical and psychiatric conditions (e.g. obesity, arthrosis, depression, anxiety), aerobic exercise, and quality of sleep. Neglecting to control for the effects of these variables increases the heterogeneity in the results. We just finalized a meta-regression and group-meta-analysis and the findings indicate that those background variables significantly increase heterogeneity.
Is smoking/not smoking relationship observed in current markers similar to other observations with these markers and smoking?
#ANSWER: As shown in Table 1 we did not find differences in smoking between fibromyalgia and patients and controls. Moreover, inclusion of smoking as covariate in GLM analysis showed no significant effect on the biomarkers. , as stated on lines 125-129: “There were no significant effects of smoking (yes/no) on the five immune biomarkers, either in a multivariate GLM analysis (F=0.77, df=4/87, p=0.55) or tests for between-subject effects (all p>0.20 even without p-correction for FDR”. We have now added a further clarification in the text (lines 138-139): “Adding smoking as a covariate (forced entry) showed that there was no significant effect of smoking on the outcome of these regression analyses.”
Fig 3. Please list SUC value as part of the name on the legend for each item.
#ANSWER: We have shown the AUC value in the caption just after AUC receiving operating curve. It reads (line 203-204): “Figure 3. Area under the receiving operating curve (0.91) of the Neural Networking model #3 with FM diagnose as output.”
Column widths in Table 2 needs to be properly adjusted. To reduce vertical spacing. Suggest decrease with of columns 1, 2,3, and column 4.
#ANSWER: Thank you for noticing this, we have fixed the column widths properly.
Please provide in, simple terms, the mathematical basis against which all final scores are produced. (this might help the understanding of the methodology for the non-machine learning readers). In figs 1, 2 & 4 the x axis "importance": pleases clarify....does this represent the weighting in the final score?
#ANSWER: We have addressed these (difficult) points by adding some explanations about how parameter weights are used to compute the relative importances. We also added a new reference (SPSS, 2019) for the interested reader. This now reads (lines 398-402): “The importance (or relative importance) reflects the strength of the relationship between the input variables and multiple interactions and the output variable, and as such indicates the predictor importance. These predictor importance values are relative because they are computed yielding a sum of 1 or 100%. These relative importances are computed based on the weights that describe the input variables as well as their multiple interactions in the neural networks (SPSS, 2019[68]).
Line 400 is too strong considering all the discussion of variation in observations of previous reports. Please consider adding to the beginning of the first sentence.... "In the observed patients, ".
#ANSWER: We have changed the first sentence of the Conclusions. It reads inline 410: In our FM patients we found lowered levels of IL-6, IL-10 and CXCL-8 independently from the effects of confounder variables
Also, please add a small statement on what a neural network is and why the use of the neural network makes sense.
#ANSWER: This statement is addressed in lines 67-70. We think that this explanation together with the response to the previous remark of this reviewer are quite adequate statements. It reads: “NN is well suited to discover more complex nonlinear relationships between input variables (i.e. cytokines, depression, anxiety) predicting FM diagnosis and widespread pain scores (the outcome variables) while examining multiple interrelated factors. Additionally, NN allows to rank the input variables in order of predictive power of the features.”
Reviewer 3 Report
The draft reports a statistics analysis using neural networks with SPSS. The reuslt is described in detail and clear. The result supports the conclusion.
Author Response
Dear Reviewer,
Thank you for reviewing our manuscript.
The draft reports a statistics analysis using neural networks with SPSS. The reuslt is described in detail and clear. The result supports the conclusion.
Include comments from this reviewer despite no answer is necessary